# Temperance, Humility and Hospitality: Three Virtues for the Anthropocene Moment?

**Jean-Philippe Pierron**

UMR Laboratoire Interdisciplinaire de Recherches—Sociétés, Sensibilités, Soin (LIR3S), University of Burgundy, 21000 Dijon, France; jean-philippe.pierron@u-bourgogne.fr

**Abstract:** As social and ecological transition and climate change raise issues that go far beyond individual responses, how can these challenges be balanced with ethical and political responses? This article intends to show that the strength of virtue ethics lies in the fact that it translates these abstract issues into concrete biographical events that shape lifestyles. The search for the good life in these matters then finds in temperance, humility and hospitality three virtues, private and social, to operate this translation. Humility makes explicit the deep interdependencies between the living, while temperance calls for practices that are attentive to these relationships, in the knowledge that our ways of life here have far-reaching consequences on the other side of the globe. This in turn invites us to restore hospitality to its cosmopolitical dimension.

**Keywords:** ethics; virtue; temperance; humility; hospitality; Anthropocene; ecology

## 1. Introduction

There is a long list of concrete practices that are being invented today to give body, through the body, to new ecologically sustainable ways of life that support social and ecological transition: eating less meat; eating organic; choosing a bank that supports the local economy and does not fund tax havens; adopting more sustainable modes of mobility; building shared habitats that benefit inter-age and inter-species relationships while also reducing the impact on land use; sorting and recycling waste; recycling clothes... All of these are ways of inventing more sober lifestyles and reflect a desire for an ethical life, a quest for the good life or "living well", as Aristotle puts it [1]. By reconnecting us with what is known as virtue ethics, the central challenge is this: to uncover our innate desire—a desire borne by every human being—to change the world, so as to direct this desire into exercises of the self, these being the foundation of a quest for the good life.

Today, people are attempting to achieve full coherence in their lives between what they understand about the ecological situation and what they believe to be necessary, existential choices to preserve sustainable and fair habitable conditions on Earth. But people do not always succeed in this, sometimes even exhausting themselves in a kind of militant burnout common among ecological activists. This calls into question whether this "logical coherence" (i.e., consistency between beliefs and action) is the sole criterion with which to evaluate the good life, or whether the good life can be judged in terms of a balance between a long-term goal, such as the ends sought in the quest for the good life, and the short-term goal of ordinary everyday choices in the context of constraints, which are sometimes contradictory. There is, for example, a temporal conflict between the long-term aim of reducing the use of pesticides in agriculture and the short-term obligation to increase productivity in order to repay loans. There is also a contradiction for a person who wants to reduce their carbon footprint but is forced to travel regularly by air for work.

In the quest for the good life, there is a renewed interest in "wisdoms" (i.e., folk knowledge and practice) that may challenge academic, scientific teachings. The latter are deemed to be too theoretical; thus, specific practices of the self are deployed in the aim of

living in greater harmony with nature. Thus, an ethics conceived as a type of dietetics is formed, relying on new ways of being, such as making ethical food choices or reducing one's ecological footprint. These ways of being activate practices of the self that break with ordinary social practices via diverse radical ecological alternatives within specific ecotopias. This leads us to propose that the Anthropocene moment surfaces a new *civilisation des mœurs* (civilization of morals), as Norbert Elias would say, characterized by a dialectic between ethics and *ethos* [2]. This dialectics is important because an ethics without an *ethos* would be inconsistent: if it does not translate into behaviors, then it would remain a posture of *belle-âme*. Conversely, an *ethos* without an ethics would merely represent a behavioral training in superficial eco-gestures.

In our view, the contemporary quest for the good life renews a broken link with the ancient virtue ethics that sought, in various ways, to live "according to nature" by mobilizing effective practices and exercises of the self. Here, we address the relevance of virtue ethics for the Anthropocene moment. The paper is organized into four sections. First, we will ask how virtue ethics responds to the Anthropocene moment. Second, we will see how virtue ethics can discern and support the desire for the good life in a culture of envy. Third, we will mediate between the concern for logical coherence and the biographical dimension of the ethical aim. Fourth, we will focus on three virtues (temperance, humility and hospitality) and their ethical fecundity for our solidarity and our belonging to the Earth.

## 2. What Is New in Virtue Ethics for the Anthropocene Moment?

The neutrality and impartiality of ecological knowledge as a scientific issue is matched by its ecobiographical impact on issues of existence, at the psychological level of course (as studied by ecopsychology, e.g., eco-anxiety) and also at the ethical level (operated by virtues). From this perspective, appealing to virtue ethics involves an ecological commitment which is an essential factor in the art of being oneself, also conceived as an art of attention toward our interdependence with living beings and the environment. A central idea is that an ecology addressing environments cannot forgo an inner ecology that relies on subjectivities and practices of the self in the quest for the good life.

We hypothesize that these practices of the self, in being aware of the Anthropocene moment, are much the same as those presented by Pierre Hadot in his work on the spiritual exercises of Greek ethics, in particular his work on the "conversion du souci" [conversion of attention]. He writes thus: "In principle, we give value to that which we care about. To change the object of attention is to effect a change in values and to change the direction of attention" [3]. In this context, philosophy, in its speculative and practical dimensions, is conceived as a "transformation of one's perception of the world", an effort that requires virtues in order to learn new ways of seeing the world. In the Anthropocene moment, the issue is to turn ecological information into ecobiographical events. Hadot put forward the idea of a kind of ecological and "ethical conversion" encouraged by new exercises of the self in support of social and ecological transition. It is not enough to demonstrate rationally or to deduce logically that other-than-human living beings, or even environments, are important enough for us to care about them. What is required is to practically care for them, for it is in this caring that we give them value. As Gaston Bachelard points out, "(t)o use a magnifying glass is to pay attention, but isn't paying attention in itself a magnifying glass? Attention alone is a magnifying glass" [4] (p. 20).

Care, as the primary virtue of attention, shifts the architectural lines of attention between what matters and what is secondary. This quality of attention is an ethical disposition, and is not the same as vigilance, which is an intellectual attitude. In caring for other-than-human living beings and environments, attention is mobilized intimately, and this creates an ecological conversion of attitudes that resonates with the interdependencies of the world, changing biological information into biographical events.

But how is the good life with and for others—and which others?—possible in the Anthropocene? What level of lifestyle commitment is required to support social and ecological transition in the Anthropocene moment? The unique challenge for virtue ethics

is to succeed in aligning the temporal and geological forces of the Anthropocene with the unique biographical time of being oneself and with the social time of being together. In the Anthropocene moment, the issue at stake is the *anthropos*: a specific conception of the self and the future of the self. "In my everyday striving, what kind of man or woman am I trying to be?" This question underlies the idea of the self and is central to conceiving a type of life—a good life as implied by virtue ethics. The renewed interest in virtue ethics comes after a long hiatus, which found its strongest justification in the Kantian tradition. For Kant, the search for the good life was so multifaceted and disordered that it was necessary, in order to guide one's actions, to replace it with the pursuit of a principle: a morality of duty. Thus, virtue gave way to duty [5] and the good life was replaced by the timeless, universal, but also impersonal Good.

However, our late modernity has brought to light the limits of such an ethical approach when faced with "hard cases", as Ronald Dworkin [6] would say, and has countered it with ethical pluralism. Furthermore, our post-traditional situation has replaced statutory morals that defined what which actions were appropriate with a demand for authenticity, which calls for an art of being oneself, of exercising the self. Our societies are, from an ethical point of view, mentally exhausting, precisely because it is up to each individual to work out what type of human they want to be. As Thomasset states:

> (t)he current return to virtue ethics is partly explained by this desire for a broader moral vision, which takes into account the history of the subject and the issue of education. Virtues, these inner dispositions of freedom which guide us towards the good that can be achieved, tackle these issues of learning desire, personal experience, and progression in a unique story, all while inserting the subject within an already existing tradition that aims at a common good. Today, answering the question "what should I do?" also involves (undoubtedly first and foremost) asking questions about the constitution of the subject and the construction of their identity: "What kind of person do I want to become?". [7]

The unique event in the Anthropocene moment is the fact that virtue ethics becomes an issue for each of us and no longer just for a small number of privileged philosophers and citizens. This means that everyone needs to work out what a good life implies in one's own biography and not as a general rule. But how good is good for me here and now?

## 3. Discerning the Desire for a Good Life within a Culture of Want

Virtue ethics does not enter the moral question via principles, rules or duties but via a "desire" for a good life. The long history of virtue ethics since Aristotle has already emphasized the importance of the good life, living well and the desire for a happy life:

> All art (*tekhne*) and all investigation (*methodos*) and similarly all action (*praxis*) and all preferential choice (*prohairesis*) tend towards some good, it seems. So it has been rightly stated that the Good is what all things tend towards. [1] (p. 31)

Why is it important to reconsider this simple idea in the context of environmental ethics? Because most of them intend to be non-anthropocentric. At first glance, the terms used by Aristotle seem to be at odds with the ultra-contemporary issue of the environment. However, this holds true only when accepting, without discussing them, two distinctions inherited from analytical philosophy: the distinction between aretaic, deontological and consequentialist ethics; and the distinction between anthropocentric, ecocentric and biocentric ethics. These distinctions are enlightening from a didactic point of view. But from a practical point of view, they are not easily mobilized and tend to remain a casuistry. Furthermore, ordinarily, moral life traverses and dialectizes, rather than dichotomizes, virtue and duty, situation and principle, intention and consequences. However, we will leave this issue open, even if it seems to us that practical wisdom resists an analytical division and refers instead to a rhythm that works intimately on moral life throughout a person's existence.

The contemporary culture of technical mastery and commercial domination of nature creates confusion between the desire for the good life and the desire to have. The latter is a form of anthropocentrism that exalts the self while also, and paradoxically, concealing the deep aspirations of this self. It is necessary to distinguish between *amour de soi* (self-love) and *amour propre* (self-appreciation) of the type that we find in Rousseau [8]. The widespread expansion of an extractivist culture that depletes natural resources as well as emotional resources—from the burning Earth to exhausted or burnt-out psyches—is due in large part to this confusion between desire and want. The subjects, uprooted from their desire for a good life, anaesthetized by technical mediations (from screens to the various ways of controlling and directing the world and living beings), reify their relationship with themselves, others and the environment, and find themselves alienated. A way out of this alienation would involve working on the internal consistency of the subject of the good life, on their self-capacities and capabilities, in order to develop a critical outer resistance to anything that prevents or prohibits the good life, based on a sense of what is right ethically, legally or politically. But in this context, how can a desire for the good life to be clarified and brought to the light? How do we clarify our desire for the good life so that it is critical of a deleterious anthropocentrism?

First, we need to set about discerning how the quest for the good life can be supported—or impeded. For that, it is crucial to differentiate between two types of finality: the pursuit of the good life driven by desire; and the finality of the "extractivist" socio-economic environment in which this desire unfolds. Indeed, as Paul Ricoeur comments:

> In Aristotelian ethics, it can only be a question of what's good for us. This relativity to us does not prevent it from being contained in any particular good. Rather, it is what is missing from all types of good. All ethics presupposes this non-saturable use of the predicate "good". [9]

We must focus on this non-saturable dimension of the predicate "good" in order to explain why an ethical life is both a goal and a striving of the self over the course of a lifetime, discerning between the possible forms of good in contrast with the idea that there is a void to be filled, a want to be satisfied, a saturated good presented as a market offering or an individualized notification in a consumerist ideology that pretends to "fulfill the expectations of a good life".

In stressing the tension between art, investigation, action and preferential choice, Aristotle already identified the non-saturation of the good as the central challenge of ethical discernment. Indeed, it is important to differentiate between ends relating to techniques and ends relating to action, especially because our time is so marked by an administrative colonization of the lived world, leading to confusion between needs, wants and desire. For virtue ethics, there is a critical opportunity to distinguish between human superiority and the critical and evaluative role of practices (*tekhne*). Ethical discernment is all about learning to coordinate and prioritize between the many ends pursued when we act or agitate.

Today, unlike in Aristotle's time, the need for discernment is ever-increasing because the desire for the good life (*praxis*) is dramatically disrupted by commercial and digital incentives or notifications (*tekhne*), and because economic success depends on the equivocation between the analog self and the digital self. But how does the force of the "new spirit of capitalism", as Boltanski puts it—i.e., the Capitalocene—shuffle the cards between priorities by promoting a commodification of the intimate which individualizes without individuating? What happens when this force crosses the border, not between needs (natural and necessary pleasures) and desire (non-necessary natural pleasures)—a distinction developed by the Greek philosophers—but between want and desire?

The ecological crisis is an abstract scientific fact. But it is turned into a biographical event according to continuous ethical choices. Thus, the social and ecological transition goes through us. This transition is not only intimate but also present in sober, temperate and attentive lifestyles. The latter need not be confused with the austerity of an ascetic renunciation, nor with instinctual exaltation of orgiastic excess lured by abundance. The ethical issue becomes a critical discourse of a political economy that encourages an acosmic

way of "making the world", i.e., of profiling collective attitudes and behaviors. This criticism is summed up in the slogan *moins de biens*, *plus de liens* [less goods, more connections]. It questions, discusses and disputes the type of world that invites excess and addiction to easy lifestyles based on the depredation of fossil fuels.

In the *Discours de la Méthode* (*Discourse on Method*), inspired by the Stoic tradition, emphasizing the values of a relatively peaceful life in a society marked by turbulence, Descartes gave himself a maxim of action: to "endeavour always to conquer myself rather than fortune, and to change my desires rather than the order of the world" [10]. This demanding proposal needs to be kept in mind by learning to distinguish between when we are right and when we misconceive our capacity to influence what is not within our reach. However, it is based on an insular conception of the self, neither porous nor buffered, and not on its relational interdependencies with its living environment. It is politically very prudent as it aims at a reform of the self, but not a reform of the world as it is. Today, the stakes may have shifted for us. We are in an age where the new spirit of capitalism is characterized by the capacity to absorb anything opposed to the market and make it the object of a new market. The search for well-being, for gentler ways of life, the quest for proximity to nature and the desire to find oases of deceleration in which to slow down in a society marked by exhausting speed seem to contradict the market society. And yet, these are also very marketable in an economy of attention. Therefore, should we not take stock of our desire to be, in order to resist the tyranny of these lifestyles which contribute to instilling in us a confusion between the desire to be and the want to have? Is it not time to identify where our desire lies in order to reform the world as it is, especially its culture of excess?

The ability to differentiate between our needs, our desires and our wants is a very powerful critical device. It involves an epistemic reconquest of lifestyles which otherwise maintain a confusion between want and desire and stop humans from distinguishing between illusions and what they really desire. While the need to eat, drink or sleep is an objective fact which can be easily identified—although it is often distorted in advertising—the distinction between want and desire is less clear. This is because the commodification of the intimate by the market fosters confusion and disorder. Is having a want synonymous to having a genuine desire? Is the gap between desire and the product suggested to me by my smartphone's artificial intelligence, which takes my "tastes" into account, also a gap between the individuating and the individualized person? How can I clearly draw a line between a personalized market offering and my deep personal aspirations? How far can we resist all the suggestions that channel our attention and make us want to have things that we do not truly desire? To discern one's desire is to work on one's internal consistency in order to develop an outer resistance to the alienating, reifying and ecologically unsustainable ways in which our societies operate, because desire is not a kind of void to be filled but a powerful call to exist. In contrast, want leads us to only one particular end: to fill a void. Desire calls us to seek experience of what really makes us tick and to unfold our own unique way of being. But confusion and trickery reduce desire to want and redirect it into its most passive form, that of envy, dependent on comparison with what other people have. Therefore, it is vital to challenge the advertising, managerial and digital devices which control and direct our desires and prevent us from experiencing the "time of desire" [11]. The time of desire is a school of freedom, so it will not be without liberation for each and every one of us.

Ethical discernment between need, want and desire to achieve a good life unfolds over the course of a lifetime. According to Aristotle, the repetitions and routines that make up the rhythms of our lives tend to establish in us a "second nature". The kind of lifestyle that is called for in a time of social and ecological transition is an ecobiographical issue. In the course of a lifetime aiming at a good life, one learns to make use, in one's own way of acting, of what has been understood about the Anthropocene moment. It conditions the future of a self expanded by the awareness of its interdependencies with human and "more-than-human" or "other-than-human" beings, and begins to ask how these may be

recognized as valid interlocutors with whom humans can co-exist. In this way, a first step is to refuse the expression "non-human living beings", which erases the plethoric diversity of life in its singularity, and denies the identities of the other.

The strength of virtue ethics therefore lies in its continuist approach to existence over the course of a lifetime even though, in the context of ecological crisis, we also need to be aware of a "threatened future", according to Hans Jonas' ethics of the future [12]. While the morality of duty is focused on the daily conflict of duty, virtue ethics sets this drama within the continuous course of a life, with all of its tensions and contradictions. Morality is discontinuous; virtue ethics is continuous. It is due to the persistent obstinacy of the virtues, which embody attention and vigilance over the course of time, that a socially and ecologically sustainable way of acting is clarified and grounded.

## 4. Logical Coherence and Biographical Obstinacy

Insisting on the long term of a modality that develops over time invites dialectization. How do the logical concern for internal coherence and persisting ethical obstinacy come together in terms of ethics? To start with, does ethical determination call for other resources than the sole formal criterion of logical coherence? Thus, one of the ethical challenges raised by the Anthropocene moment raises is the concurrence and discordance between timescales, the emergency of climate change and of an ecological crisis and the short length of a life. To respond to this challenge, one may claim the demand for logical coherence as the (sole) criterion and summit of moral life. Disregarding how this plays out over time, this claim imposes the achronicity of the logical decree onto the temporal dimension of ethics. An incoherent good life would be inconsistent. This idea translates into a radical, demanding and generous call for the synchronization of words and actions, theory and practice and ecologically ethical thought and the personal ecology of such thought. Indeed, how can one grasp the earnestness of social and ecological transition without achieving coherence with, and entering into, that transition? The ways of being oneself mentioned earlier become expressions of the self calling for coherence—e.g., deciding not to travel by aeroplane because of $CO_2$ emissions; reducing meat consumption; refusing to use laptops made by big extractive companies, in order to resist a consumption of the world which is also a consumption of the self; or distancing oneself from the logics of the market and its normalization.

But these radical ways of life demand profound self-reform by seeking logical coherence and are simultaneously stimulating, energizing and disturbing: stimulating, because they show that it is possible to initiate radically new ways of life, where it may previously have been thought impossible. They demonstrate a form of practical inventiveness and a salutary and promising poetic and ethical innovation. Energizing, because the testimonial scope of these ways of life set in motion attests to the viable and desirable nature of such life choices. Disturbing, too, in two ways: they disturb individuals by drawing them out of their comfort zone so as to take part in the transition, and they threaten those who refuse this logical coherence out of indifference or selfishness. In the name of coherence, one may refuse any ethical compromise on the basis that to accept a compromise would be to compromise oneself. In this context, one can exhaust oneself in trying to achieve the impossible task of being coherent, an ethical exhaustion marked by the enormous discrepancy between self-reform and the gigantic powers against which one needs to fight in order to bring about change. This intransigent call for logical coherence crushes the temporal dimension of the moral life and its work of internalizing issues over the course of a lifetime. It neglects the fact that, in matters of ethics, presenting a problem from a rational and logical point of view is not the same as resolving it. Deciding to live a good life is not a logical solution but an ethical determination that commits an individual for a lifetime.

The primacy of the logical over the biographical can be enforced in the name of a very violent ethical purity conceived as logical coherence. Conversely, the purpose of ethical discernment, because of the situated nature of our ethical positions, is to distinguish between the ethical demand for a radically good life and ethical intransigence, which can

be brutal. To put it another way, the ethical challenge consists in asking oneself how to be radical without being marginal.

Alasdair MacIntyre insists that lives and ethical self-narratives do not exist in a vacuum [13]. They unfold in the context of belonging to living environments which oppose our "aspirations to be" with other competing, dominant, supporting or contradictory narratives. Discernment and ethical deliberation do not operate as logical deductions, even if the rigorous path of reasoning and the orderliness linked to coherence appear prestigious and refined. MacIntyre prevents the temptation to lapse into ethical solipsism by highlighting that a moral subject is constructed in connection to the traditions of meaning provided by living communities. He warns against overvaluing the criterion of logical coherence as the sole and definitive criterion of the good life.

Authentic ethical conduct and moral judgement become defined and formulated in and due to learning the practices, *ethos* and habits of a given historical community, including ways of relating to other-than-human beings and to the environment. This narrative approach to human identity reiterates that identity is constructed and recounted via a self-narrative, which may be a counter-narrative to the major dominant ideological narratives. This recognition could illustrate what history emphasizes and MacIntyre questions: "I cannot answer the question, 'what should I do?'" he said, "until I answer the question that precedes it: 'Of what story or stories do I find myself a part?'" Achieving a good, meaningful and unified life takes time. This unity is not given but conquered. It is developed in a way that one situates oneself with regard to social practices, inscribing one's life story as part of a living tradition, via practical inventiveness: "I cannot answer the question, 'what should I do?'" he said, "until I answer the question that precedes it: 'what histories am I part of?'" [13] (p. 210).

For Anthropocene biographies, it will therefore be a question of breaking away from the major dominant narratives (economic growth, the market, nature) and inventing other minor narratives, other metaphors. It is important to learn to discern in order to position oneself among the trials of friction, tension and equivocality with which one must live. The human world is multifaceted, and it would be an illusion to believe that the great clarity of the criterion of logical coherence could on its own eliminate the equivocality and confusion which make a world of ethical action, in the world of humans and their relationships with their environment.

The world of logic cannot on its own create the logic of the ecological world. This is why there will be controversy and conflict in the interpretations provided and motivated by an ecological democracy. It will be possible to oppose the ordinary pace of the world, in various ways, in various types of ethical life. This may happen by withdrawing into oneself and forming a type of inter-self (as seen in the radicality of utopias of withdrawal in forms of autarkic community practices among neo-rural populations). This may also lead to violent rejection and even to revolution, as seen in the revolutionary radicality of utopias of protest that oppose neoliberal logics in ecotopias (e.g., the "Zone to Defend") which attempt to spatially situate struggles against neoliberalism. To be able to choose and position itself, the ethical self of ecological consciousness needs to learn, not using deduction but deliberation, to discover and ] adopt the appropriate attitudes in order to succeed in meeting the demands of that self. In the next section, we will shed light on this demand by focusing on three virtues (temperance, humility and hospitality) and reflect on the transition from self to more than self by being aware of our belonging to the Earth.

### 4.1. Three Virtues towards the Decentring of Self in the Anthropocene: Temperance, Humility and Hospitality

4.1.1. Temperance and H.-D. Thoreau

How to define sobriety and temperance with regard to the "new spirit of capitalism" [14]? Temperance is one of the cardinal virtues of Greek ethics and is often discussed in the context of virtue ethics. Within the framework of ecological transition, it is often termed "moderation", "happy sobriety" [15] or "voluntary frugality". Temperance is char-

acterized by a sense of moderation in contrast to its opposite, excess or hubris, i.e., refusing to be constrained. This virtue is particularly relevant in societies marked by abundance, excess, food wastage, etc., where the excess of intemperance seems to be the baseline of ordinary action. Today, hubris is encouraged by the technological solutionism of transhumanism and can be seen in the never-ending economic activity of Western society: infinite growth in a finite world supported by unbridled economic growth based on a cult of want.

*Sōphrosunē*, the Greek word for temperance, means "to have one's whole mind about oneself", or, in other words, to have a level of self-knowledge that, in turn, enables self-control. For rational people who do not take themselves for gods, excess or hubris can be seen as a pathology: losing one's mindful awareness concerning oneself, blurring the boundaries between the mortal world and the realm of divinity. Today, in our secularized societies, where religion no longer imposes rules and transgressions are no longer sacrilege, any limitation in the name of temperance can only be a self-imposition. The issue is not to "live according to nature" conceived as a cosmos since this would represent only an order for things. It is about rediscovering an intimate sense of moderation as something desirable in an authentic experience of the self.

In the next section, we will shed light on this demand by focusing on three virtues (temperance, humility and hospitality) and reflect on the transition from self to more than self by being aware of our belonging to the Earth.

For Henry-David Thoreau, one figurehead of ecological thinking, an intimate sense of moderation is characterized by two traits: a form of self-knowledge which preserves the wild part of oneself; and performing exercises of the self in the practice of voluntary simplicity and sobriety [16]. The first of these traits is characterized by understanding and self-concern about what it really means to live in meditation and solitude. The thrust of this trait concerns the experience of the wilderness as a test of solitude. It is not an experience of isolation or loneliness but rather a condition of deepening oneself and a revery in communion with all beings, both human and other-than-human, who populate the world and bring it to life. With regard to the stimulation of technical societies, the practice of solitude and moderation is not about disdaining these stimulations: it is about mastering them. The issue is to be aware of the often sterilizing norms of social life, and therefore, a contrario, to strive to sensitize ourselves using moderation and to replenish our imagination and our inner life using soothing or invigorating physical contact with the elements. The self-disposition of moderation can be stimulated using specific ecotopic devices such as "Operation Walden". Ecotopia can be related to what Foucault, in line with Bachelard in *La poétique de l'espace* (*The Poetics of Space*), designated a "heterotopia", a proposal for a liberating "counter-location". Eco-heterotopia opposes the negative impact of modern society's delineated spaces of control. It also aims to counter the worldly solicitations that subject our bodies and lives to the injunctions of the market and, now, to digital hyper-connection—which is another type of biopolitics. Bachelard, a learned reader of Thoreau, the philosopher in the woods, invited us to live and dream the experience of solitude as that of an experience lived in the full presence of our earthly dwelling. The etymology of the verb "to dwell" embodies the profound assurance of our "being there", of our "ecological self". While the ecological crisis can be seen as a crisis of dwelling, the experience of moderation found in the solitude of Walden's hut invites us to recover an intensity of presence in the joy of complicity.

> The hut is centred solitude. (. . .) The hut cannot receive any of the wealth 'of this world'. It has a happy intensity of poverty. The hermit's hut is a glory of poverty. From one stripping to the next, it grants us access to absolute refuge. [4] (p. 88)

The stripping away of temperance is not impoverishment but a pruning that brings new life.

The second trait translates the call for moderation into a call for a simple life and implies a type of ethical self-reform. The desire for simplification and moderation targets the excesses of modern societies in terms of luxury and excessive consumption. These excesses obliterate the self since they create a confusion between self-love (which takes

authentic account of one's true desires) and self-appreciation (a biased look at oneself mediated by the mirrors constructed by societies)—a confusion pointed out by Rousseau (see above). In *Walden ou la vie dans les bois* (*Walden*), Thoreau exemplifies the prototype of what is being sought today in terms of temperance. An experience of the self leads to freeing oneself from the masks and ambiguities caused by the commodification of the intimate and the confusion between desire and want. This perspective has inspired an appreciation of the "wilderness" in environmental ethics. One can clearly see here that care for the "wild" concerns both the protection of the wild fauna and flora outside of oneself (the outer wilderness) and desire, which is the wild part of the self inside oneself: the "inner wilderness". Thus, Thoreau writes,

> Before arranging our homes with objects that we find beautiful, we must strip the walls, just as we must strip our lives.

> Simplicity, simplicity, simplicity! Let your belongings, I tell you, number only two or three, and not one hundred or one thousand (…). Simplify, simplify. Instead of three meals a day, if necessary, have only one; instead of a hundred dishes, five; and reduce the rest accordingly. [17] (pp. 121–197)

Economy is far from being a secondary concern for Thoreau. By giving the heading *Economy* to the first chapter of *Walden*, he rearticulates economics and ecology using an ethical minimalism. Moderation is essential to living well, freeing oneself, if not extricating oneself, from worldly affairs with their superfluity, their luxury and the call for self-diversion. Thus, it is in the here and now of the situation that the ethical elevation of the subject takes place. Our first waste is always a waste of life. The virtue of temperance therefore operates a reversal, an ecological conversion, of the meaning and value of the experience of responsible and equitable consumption. Perceived as economic poverty by those who feel excluded from the market system, it becomes ethical simplicity for those who transvalue its meaning. Moderation or sobriety are not ways of escaping in order to forget oneself, but on the contrary, experiences of a deepening of the self. Whereas civilization distances us from ourselves by interposing its veils, virtue is a return *into* oneself, which is a return *to* oneself.

> I would go into the woods because I wanted to live deliberately and face up to the essential facts of life, to find out what it had to teach me, so that I wouldn't feel, at the time of my death, that I had not lived. [17] (pp. 195–197)

With moderation, temperance simplifies ordinary life in order to intensify life in one's desire to be. Striving to eliminate the superfluous and to have less, of letting one's actions be guided by calculations of ecological or carbon footprint (e.g., with reference to a climate map), is not quantitative. It is qualitative: not living less but living better by changing the meaning of what it is to live. Whereas the culture of want sees all sobriety as a loss, as less, as a devaluation, the culture of desire in the simplification of moderation aims for a more intense life. The care of desire in moderation as a remedy for ecological crises is not an austere cure of abstinence. By intensifying our relationships with others and with our living environment, it is more a matter of desiring better than desiring less.

### 4.1.2. Humility and A. Leopold

Is ecologism an antihumanism, a human humiliation or the best way to understand humans? Humility is a virtue that attempts to understand rightly who we are and that resists any form of excess or hubris. The idea of humility is close to an understanding of what Arne Naess called the "ecological self", as a form of otherness in oneself which is greater than the self [18]. The ecological self goes beyond the limits of the individuality (ego) of our socio-political affiliations and is open to our interdependencies with other-than-human beings and environments. The emphasis in contemporary ecological thought on the idea of the human as an "earthling" or "terrestrial" being is echoed by the etymological link between humanitas and humus in Latin. The same Latin root is shared by the words "humus", referring to the soil and the "living quality of the soil"; "humanity", understood

in terms of its links to the Earth and "humility", which sees these links as interdependencies and not as alienations. Accordingly, humility is the ecological virtue that responds to the excessiveness of our uprooting from the Earth, characterized by a culture of domination and extraction. Humility rightly resists a position of domination and responds to it by adopting a more modest position in which one is aware of one's incompleteness or insufficiency.

One should not confuse humility with its caricature, humiliation. The social and ecological transition does not aim to inflict a humiliation on humanity. By calling a form of humility, it seeks to give back to humans their right place, a place that is neither excessive nor ridiculous. Humility is not the experience of abasement or inferiority. It is daring to stop thinking of and envisaging oneself in terms of superiority. Only then can humility serve as a critique of anthropocentrism—and not by fostering a kind of hatred of humans. To think better, hence in humility, of humans is not to think less of them in a form of humiliation. Along this line, in *L'Almanach d'un comté des sables* (*A Sand County Almanach*), Aldo Leopold formulated a very humble call to "learn to think like a mountain". This call implies that one necessarily undergoes an inner/internal displacement, a decentering of oneself. This leads to seeing oneself, in one's own way of making the world, from the perspective of a mountain. One needs to consider our ways of belonging to the Earth since it began, not from the point of view of our anthropic domination. To this end, humble practices need to be exercised.

For example, how do we address the fields of architecture and urbanism whose extractivist practices in the search for materials and the establishment of building sites have a major impact on social and ecological systems, and how could they be made more humble? We are not unaware of the excessiveness of the mega-towers and the technological solutionism which make the gray cement of our cities and depletes resources. Neither do we ignore the prestigious unilateral gesture of the architect's signature responding to this uninhibited anthropocentrism. But the transition from urban development practices to those of resource management by recycling should put the emphasis on relationships and life habits, promoting space maintenance and renovation practices. This opens up humble options to take care of ecosystems and to increase the sustainability of places and environments. Along this line, we may consider other rural, peasant and indigenous lifestyles around the world. The lifestyles preponderant in urban cultures are unknown in other cultures around the world. To recognize diverse cosmovisions of the "good life", e.g., the "buenos vivires" of Andean and Amazonian people, could be epistemologically crucial to shift the way architecture and urbanism are promoted.

> For the urban factory, the world. . .is not a generic space, but a preliminary and inextricable fabric of intertwined vital forms. . . .In view of all these interwoven dynamics, it is clearly better to maintain and care for [these forms], rather than decomposing and recomposing [them]. [19]

Without feeling diminished, the builder or architect can become the one who joins together and works with those relationships (both human and other-than-human) that they know, that they can listen to. Decisions about which projects to work on, the choice of construction techniques and the use of recycled materials (industrial ecology) or materials with short production chains—or, to the contrary, the use of harmful extractivist practices (involving, for example, sand and cement)—engender a process by which the professional identity of an architect or urban planner is shaped in continuous choices which are never neutral. The shaping of the self reverberates on the architecture, because the sustainability of the materials dictates the duration of the projects. Building gives a "semblance of eternity to the fragility of human affairs" says Arendt, since it sketches a very specific and stabilized type of common world [20]. It transcribes into materials what enables and nourishes relationships between humans, more-than-humans and environments, or which can, to the contrary, destroy or prevent those relationships—e.g., the impermeable city which addresses water as a problem versus the "sponge city" which creates relationships with water.

### 4.1.3. Hospitality and Climate Migrants

How can we guide globalization ethically and politically so that it takes care of nature and humans, that is, how to exert the virtue of hospitality? Virtue ethics applied to ecology could certainly provide a fruitful context for thinking about care of the self and supporting a form of self-esteem. But the centering on the self may be seen as an ethical comfort and a form of self-centered—if not narcissistic—complacency. The concern of achieving a good life for oneself, leaving aside the affairs of the world, its possible collapse and extinction, may be a complacent form of enjoyment of one's self-assured inner citadel. How do we avoid ethical self-concern that leads us to turn only to ourselves? What might be the political implications of promoting virtues while their contribution to ecological transition may seem marginal? Although they may raise concern for beings other than oneself, including other-than-humans, the latter will never be visages ("faces"), to use Levinas's term [21]. How good, in short, are virtues that concern individuals, but not institutions, local authorities or States?

In response to these criticisms, we point to a virtue which lies at the border between ethics and politics, and has gained central importance as part of the new cosmopolitanism called for by the Anthropocene moment. This is the virtue of hospitality. Following in the wake of MacIntyre, the philosopher Alain Thomasset defines hospitality as a "social virtue":

> By social virtues, I mean the virtues that are at stake in relationships with others, and more specifically, in the functioning of society as a whole. The virtues, in fact, do not concern only the conformation of an individual to a personal morality (for example in their family life), but also the behaviour of every person in their contribution to common life. [22]

In line with one's concern for a common world, social virtues such as justice, solidarity and hospitality, therefore, are to be found in the space between the self and the more than self, between oneself and others (human and other-than-human). They sustain the pursuit of a good life as the long-term goal of our ways of making the world, which need to be translated into socially and ecologically sustainable institutions within the common world.

How does the virtue of hospitality fit into a framework of ecological thought? Hospitality meets what Kant called the "sphericity" of the Earth and the global mobility it calls for. This sphericity has generated a globalization of our technical systems and also of economic exchanges. Global humanization is an ethical issue. How can we guide globalization ethically and politically so that it takes care of nature and humans? How can the goals of a good life lived with and for all beings be added from the outside to a culture that controls and directs our relationships with nature? Like it or not, our global, technical and economic interdependencies are such that we are contemporaries of all human and more-than-human beings on the planet. Thus, our excessive ways of life are too often linked to impoverished ways of life elsewhere.

Hospitality resonates in the space between the self and the more-than-self due to the two-sided meaning of its root name hostis: hospitality and hostility toward the other. We believe that hospitality will probably be challenged when confronted with those who are forced to migrate from inhabitable life environments, as this will likely disturb and destabilize our lifestyles, which so heavily depend on the resources of others. Such is the challenge of hospitality. Openness to alterity can alter us. The arrival of a stranger can prompt in us hostilities that disturb us, especially when we are open to welcoming that stranger. The ethical challenge will be to ask why others are forced, as refugees or migrants, to leave their homelands and in what ways our lifestyles contribute to this.

The ethics of hospitality contributes to reterritorializing within our lives migratory issues that seem pre-destined or out of our control. Indeed, the loss of people's home may be in part caused by us—whether they no longer have a home (landless peasants) or their land has been devastated, is submerged under rising waters or has become unfertile. This presupposes, first of all, that we are able to see in the foreigner that we too are a foreigner and also a fellow human on Earth. Hospitality, as an ethical virtue, reminds us that we

are, on some level, a guest passing through even in the very place that we believe to be our own. This reminder links together hospitality and humility. It helps us to remember that we are the foreigner who at some point has been the recipient of hospitality offered by others and by the Earth. The massive international mobilities inherent to forced migrations pose an ethical and political challenge concerning the sustainability of our ways of making the world. Many of the military conflicts that have led to the displacement of people have ecological causes: water wars in particular. It will be necessary to move from the de facto solidarity of all humanity that arises from shared problems (the Anthropocene, climate change, species extinctions, erosion) to a deliberate solidarity that gives ethical and political significance to an awareness of our mutual interdependencies. The virtue of hospitality becomes the figure of a new cosmopolitanism in a post-Kantian sense as it recognizes the other in its standing as an earthly being, as a "citizen of the world". On a territorial scale, it opposes the practices of an inhospitable necro-capitalism that generates environmental and social injustices in the destruction of habitats. It also responds, at a global scale, to the ecosystemic effects of climate change, such as the impact of ecological disasters on social environments, generating mass migration and climate refugees. Yet, what attitude should we adopt toward the "other" who migrates? This question is both intimate—as it refers to a personal invitation extended to someone nearby—and at the same time very public, hence challenging a State, or even a continent, in terms of how it conceives of hospitable cosmopolitics. Hospitality is a virtue that makes it possible to consider the arrival of people from climatic migrations in a different light, clearing a path between welcome and worry, availability and mistrust, the ethical generosity of openness and a realpolitik that asserts that a "State cannot accommodate all the misery of the world" [23]. To be hospitable means to experience an amazement at the life journey of a migrant who has come to us (e.g., by crossing the Mediterranean sea or the entire South American continent) so as to act hospitably. The actions of charities that welcome migrants are all, in different ways, examples of the virtue of hospitality, see Cimade (https://www.lacimade.org, accessed on 5 December 2023), the Welcome Network (https://www.jrsfrance.org, accessed on 5 December 2023), L'auberge des Migrants à Calais (https://www.laubergedesmigrants.fr, accessed on 5 December 2023), etc.

One of the challenges is to understand the issue of hospitality for climate migrants based neither on a vague, general or administrative idea of migrants—of the kind that would separate them into categories like "political", "economic", "social", etc.—nor on a form of compassion fatigue. The virtue of hospitality fights against the inhospitable stereotypes about migrants by daring to open up to the call of the other. It keeps in mind the fact that behind the ready-made and generalizing imagery of the "Migrant" or "Foreigner", there is a unique life story that has unfolded against the backdrop of a socio-ecological disaster. Hospitality questions how a society treats the lives of those that it places in its margins, in the land of exile. The virtue of hospitality works to document these lives, which have been hidden by public policies shirking responsibility, and this documentation in itself is a form of welcome. By fostering exposure to the other, hospitality fights against a culture of collective anesthesia, testifying precarious lives and the conditions that are created for them. In this, this virtue of hospitality discovers its full ethical potential as a critique of migration policies. This is shown by analyzing the commitment of volunteers who, seeing that such policies are placing migrants in danger, choose to rescue them at the border. The sociologist Anne-Claire Defossez studied this in the ethical and political clashes at the Italian–French border in Briançon in 2020 [24]. These clashes were in stark contrast to the hospitality of volunteers providing first aid to migrants harmed by the cold. They revealed the "realism" of the police force who prevented the volunteers from this action, in order to avoid setting a "precedent" that might attract more migrants in the future. The border is where the tension between hostility and hospitality materializes. There, the ethical dilemma between turning migrants away or rescuing them is developed, experienced and established, pushing "nationals" to oppose "foreigners", toward whom empathy is

prevented or even prohibited. In this context, the hospitality shown by volunteers is treated by the police and far-right activists in the same way as the practices of smugglers.

There is no hospitality without concrete practices of hospitality by which—necessarily—one can be affected, disturbed and changed by the other in order to also ethically guide and support the ecological transition.

The migration histories put together by charities to help with asylum applications is both a translation exercise and an exercise of hospitality, and brings into focus the critical and political nature of this virtue. The use of subjective and itinerary maps [25], which allow oneself to be affected by the other's story, embodies, within the context of a narrative ethic, the hospitality of listening. The maps help to recover a life story that cannot be recounted in terms of measurable displacements. This exercise of hospitality aims to strip away stereotypes in order to discover incredible, and often horrifying, life stories. The political virtue of hospitality therefore resists clichéd ways of speaking about influxes of migrants and opposes ideologies that stir up abstract imageries of invasion or widespread replacement by "hordes of climatic migrants", to instead pinpoint and question the ecological logic, the deadly social practices and ecocides at the root of this mobility. The virtue of hospitality is an opportunity for everyone to remember having once been a foreigner subject to displacement. It maintains a *visage* (face) of exteriority that resists any institutional or political ideology in which the history of the other is instrumentalized [21]. The virtue of hospitality reveals what is specific, where ideology generalizes.

> When we show our care by welcoming them, we can in return gain a better understanding of their situation, of the causes of their distress and of their unique experience. And then the movement is reversed again: the recognition that we are able to give becomes a new gift, more profound than that of food or shelter. Hospitality to strangers, the needy and the poor, gives us a direct emotional contact that deepens our understanding of the social changes needed and inspires actions of solidarity that can lead to a global transformation. When we welcome the foreigner, we are invited to discover other worlds. The initial animosity becomes a fruitful friendship. [26]

A social virtue, hospitality maintains the ethical demand for welcome at the heart of the constrained, pragmatic, supposedly realistic and often cynical demands of the "ethics of responsibility". Remembering our history as human migrants temporarily passing through the Earth humanizes us. It invites public policies to protect the rights of refugees and migrants. Human rights are also social and environmental rights shared by all humans who are inhabitants of the Earth.

## 5. Conclusions

Putting into practice the virtues of temperance, humility and hospitality helps us to discern between need, want and desire; between consumerist excess and moderation; between a healthy self-esteem and an anthropocentric self-exaltation and between fear and vulnerability in hospitality. This discernment contributes to the process of shifting from a lifestyle centered on the self to one that takes care of and even changes the world. The virtues of temperance, humility and hospitality are not the only ones that need to be mobilized in the process. There are other virtues: justice, attention, solidarity, love, friendship, etc., all of which help us to develop discernment in our complex and confusing way of life. In this paper, we have focused on these three abovementioned virtues because, first, they allow us to move from the personal to the public dimension, and together they form a system. Humility clarifies our understanding of deep interdependencies; it invites temperate practices that practically and symbolically connect our ways of life with distant lives who may be forced to migrate toward us and that appeal to our hospitality. According to Donna Harraway, the ethical existence of multi-specific and multi-continent biographies in the Anthropocene is initiated within this network of virtues [27]. We recognize that our continued existence is possible only thanks to all the human and more-than-human beings in the world.

Virtue ethics in emphasizing the existential attitudes of individuals helps to raise attention to life stories in well-defined and specific situations and to make sure that debates on social and ecological transition do not remain purely abstract.

Virtues call for a personal, intimate appropriation of what *affects* and *effects* them so that the inner self can be transformed. Because of its disparity with human capacities, the global dimension of planetary forces may encourage a new destiny. Virtue ethics responds by addressing planetary issues in the here and now of a life. From within a fleshed-out, ethical context, it approaches the Anthropocene moment, not in abstracto, but by considering the effects on lives in terms of their environments or habitats, their minds and their relationships. The fecundity of virtue ethics lies in its critical and practical implication to support the ecological self and guide the social and ecological transition. In a culture of the self and exercises of the self, it offers an internal *consistency* understood as being propaedeutic to an outer *resistance*. It is complementary to other ethical approaches that are mostly focused on mitigating or preventing ecological disasters. But much of this disorder will not be prevented and it is crucial to promote virtue ethics, e.g., in education, so that human beings develop the capacity to respond to disturbing events and integrate them into their biographies, their ecobiographies [28].

**Funding:** This research received no external funding.

**Institutional Review Board Statement:** Not applicable.

**Informed Consent Statement:** Not applicable.

**Data Availability Statement:** The study did not report any data.

**Conflicts of Interest:** The author declares no conflict of interest.

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
