# Peer review of "Temperance, Humility and Hospitality: Three Virtues for the Anthropocene Moment?"

_philosophies, doi:10.3390/philosophies9010005_

Round 1

Reviewer 1 Report

Comments and Suggestions for Authors

Authors propose, from the virtue ethics of Aristoteles, three key virtues in the face of the (historical) moment of the Anthropocene that is being experienced globally. Virtues as, temperance, humility and hospitality generate actions in the public space beyond personal. These virtues would provide meaning to lifestyle changes promoted in the current discourse of the transition to sustainability.

The article is clear, and the issue very relevant to inquire about human being role in the multiple crisis scenarios. My main concerns are:

1. Authors use "non-human living beings" to name the plethoric diversity of life. However, naming the other, identifying them as another with non-human qualities rather than for their singular qualities is biased to the anthropocentric view of the world. In the environmental ethic literature is used "more-than-human", also "other-than-human". I invite to authors think about it. For example, see Rozzi R (2018) Biocultural homogenization: A wicked problem in the Anthropocene. In: From biocultural homogenization to biocultural conservation. Springer, Cham, pp 21–48

2. The opulent lifestyles associated with urbanization may ignore other rural, peasant and indigenous lifestyles around the world. These lifestyles preponderant in urban cultures unknown other cultures around the world. Recognize diverse cosmovisions around “good life” especially in the sense of “buenos vivires” of Andean and Amazonian people could be epistemologically just to the broad thought around this topic among diverse intellectual traditions.

Also, this recognition could illustrate the history emphasize and the MacIntyre questions (L265, 266): “I cannot answer the question, ‘what should I do?’” he said, “until I answer the question that precedes 265 it: ‘what histories am I part of?’”

In this way, the article will not ignore others visions and historical position around global crisis (as those associated with the Anthropocene moment).

3. In front of “confusion between need, want and desire”, and the conception of philosophy as a “transformation of one’s perception of the world” (L63). I invite you to reflect about how philosophy could work in interdisciplinary fields to promote discernment think to contribute to “engages the virtues in order to learn new ways of seeing the world” (L64).

4. If you consider important, I suggest you retake the idea about the “spiritual exercises of Greek ethics”, that you introduced at the beginning but forgotten at the end of manuscript. I guess that this practice could be very interesting to discuss about the limitations in achieve the lifestyle changes. Especially if you emphasize the philosophy role in an interdisciplinary field to “engages the virtues in order to learn new ways of seeing the world” (L64).

Minor comments:

L120. You lost the final point.

L209-211. Beautiful!!  For this reason, we need learn to see others in themselves, from their singularities, not from the conception of being non-human that denies the identity of the other.

“It questions the future of a self expanded by the awareness of its interdependencies with human and non-human living beings, and begins to ask how these may be recognized as valid interlocutors with which humans can co-exist.”

L410 This section for the hospitality virtue is not highlighted in bold. See L286 for temperance, L323 to humility.

Author Response

dear colleague

thank you for taking the time to read my contribution and for these useful comments.
Concerning the naming of non-humans, we have the same controversy in France. I agree with you and I will correct it accordingly. I agree with your observations about the lifestyle of non-urban environments and other cultures.
I'll try to find a relevant answer to point 3 concerning collaboration between philosophy and other forms of knowledge.
Your suggestion for point 4 is a good idea and will unify my comments.
Kind regards

Reviewer 2 Report

Comments and Suggestions for Authors

This paper raises a number of interesting ideas related to a virtue ethical response to environmental concerns. There are some strengths, in particular, the discussion on the three virtues that the authors claim can translate abstract ethical and political environmental concerns into “concrete biographical events involving lifestyles”. The three virtues highlighted seem apt for the role assigned to them, that is to provide individuals with “practices of the self” that help them to translate ethical and political environmental concerns into concrete practices in the context of the “Anthropocene moment”. 

The discussions of the virtue of temperance in light of Thoreau’s life of simplicity outlined in “On Walden Pond”, the virtue of humility in relation to Leopold’s A Sand County Almanac, and the virtue of hospitality in the face of impending climate migrations are both interesting and suggestive. 

That said, the paper has crucial flaws that should be remedied prior to being considered for publication. First, the paper is poorly organised both as a whole and in relation to the parts. This is especially evident in the introductory sections which are prolix and lack a clear point. The divisions between paragraphs seem arbitrary, rather than marking the introduction of a new part of the argument. Second, and relatedly, while the writing is expressive and evocative it is untargeted, and replete with questions both sincere and rhetorical. The questions are often asked mid-paragraph and many of them seem so poorly conceived as to be utterly unanswerable (for instance “How do I clearly draw the line between a personalized market offering and my deep personal aspirations?” and “should we not take stock of our desire to be in order to resist the tyranny of these lifestyles which contribute to installing in us a confusion between desire to be and desire to have?). Finally, the paper contains many references to philosophers and their work, but it is entirely uncritical. Authors are mentioned but their work is not explained or analysed with any consistency, nor is there evidence critical engagement with the claims discussed. 

In sum, the paper requires significant rewriting so that it is a good degree clearer and more succinct. Only then might the many philosophical claims which are suggested and asserted, rather than clearly argued for, be adequately assessed.

Comments on the Quality of English Language

Please see above

Author Response

dear colleague
thank you for taking the time to read my contribution.
I will give some thought to your comment about dividing up the paragraphs to make it easier to read.
Concerning the references to philosophers, I understand your comment but my aim in this article is not to discuss philosophers in a critical way. Rather, they are used here as guidelines for developing virtue ethics in an ecological context.

As for the style of my comments, which I consider too rhetorical, I recognise that I have deliberately chosen to write in a way that is broadly questioning. It's not a question of sophistry, or even rhetoric, but of working on the very language of ethical questioning. Perhaps the translation from French into English blurs this project a little?

thank you for your comments 

Round 2

Reviewer 2 Report

Comments and Suggestions for Authors

Neither of the two documents provided are not in a fit state to be published. One still has the comment threads from an external editor, and makes no reference to the previous peers review comments. The other has resolved the comment threads but in such a way that it includes in the text instances of the comments made by the editor in French. The manuscript cannot be accepted for publication in this state.

Comments on the Quality of English Language

The manuscript has not been properly edited. It contains irrelevant comments in French. 

Author Response

as recommanded by academics editor, I've provided references for each authors mentioned in the text; move informations in the text in footnotes; simplified the phrases and checked typos
